# Fermentation Pattern of Several Carbohydrate Sources Incubated in An in Vitro Semicontinuous System with Inocula From Ruminants Given Either Forage or Concentrate-Based Diets

**DOI:** 10.3390/ani10020261

**Published:** 2020-02-06

**Authors:** Zahia Amanzougarene, Susana Yuste, Manuel Fondevila

**Affiliations:** Departamento de Producción Animal y Ciencia de los Alimentos, Instituto Agroalimentario de Aragón (IA2), Universidad de Zaragoza-CITA, M. Servet 177, 50013 Zaragoza, Spain; zahiaagro@yahoo.fr (Z.A.); susana_monre@hotmail.com (S.Y.)

**Keywords:** cereals, fibrous byproducts, gas volume, pH, volatile fatty acids, in vitro fermentation

## Abstract

**Simple Summary:**

A sudden change from a milk/forage diet to a high concentrate diet in young ruminants increases the rate and extent of rumen microbial fermentation, leading to digestive problems, such as acidosis. The magnitude of this effect depends on the nature of the ingredients. Six carbohydrate sources were tested: three cereal grains (barley, maize and brown sorghum), as high starch sources of different availability, and three byproducts (sugarbeet pulp, citrus pulp and wheat bran), as sources of either insoluble or soluble fibre. An in vitro semicontinuous incubation system was used to compare the fermentation pattern of substrates incubated with inocula-simulating concentrate or forage diets, under the pH and liquid outflow rate conditions of intensive feeding systems. The magnitude of microbial fermentation was higher with the concentrate than the forage inoculum, and the drop in pH in the first part of incubation was more profound. Among the substrates, citrus pulp had a greater acidification potential and was fermented at a higher extent, followed by wheat bran and barley. In conclusion, the acidification capacity of substrates plays an important role in the environmental conditions, depending on the type of diet given to the ruminant. This in vitro system allows us to compare the substrates under conditions simulating high-concentrate feeding.

**Abstract:**

The fermentation pattern of several carbohydrate sources and their interaction with the nature of microbial inoculum was studied. Barley (B), maize (M), sorghum, (S), sugarbeet pulp (BP), citrus pulp (CP) and wheat bran (WB) were tested in an in vitro semicontinuous system maintaining poorly buffered conditions from 0 to 6 h, and being gradually buffered to 6.5 from 8 to 24 h to simulate the rumen pH pattern. Rumen fluid inoculum was obtained from lambs fed with either concentrate and barley straw (CI) or alfalfa hay (FI). The extent of fermentation was higher with CI than FI throughout the incubation (*p* < 0.05). Among the substrates, S, BP and M maintained the highest pH (*p* < 0.05), whereas CP recorded the lowest pH with both inocula. Similarly, CP recorded the highest gas volume throughout the incubation, followed by WB and B, and S recorded the lowest volume (*p* < 0.05). On average, the total volatile fatty acid (VFA), as well as lactic acid concentration, was higher with CP than in the other substrates (*p* < 0.05). The microbial structure was more affected by the animal donor of inoculum than by the substrate. The in vitro semicontinuous system allows for the study of the rumen environment acidification and substrate microbial fermentation under intensive feeding conditions.

## 1. Introduction

Reaching a high productive performance in the fattening of young ruminants requires high-energy diets that promote a high rate and extent of rumen microbial fermentation, with acidosis as a frequent consequence [1]. In practice, ruminants reared at pasture are often abruptly introduced to intensive feeding systems without being previously adapted to high-concentrate diets, promoting variable responses in the rate and extent of fermentation [2]. Cereals are commonly considered as ingredients of concentrate diets for ruminants. Their energy value depends on starch availability, which differs according to their chemical structure, protein matrix or, in some cases, the presence of phenolic compounds [3,4]. Fibrous byproducts with either insoluble (cellulose, hemicelluloses) or soluble (mostly pectin) polysaccharides and variable proportions of either starch or sugars [5,6] are also included among the carbohydrate sources currently used. Fitting substrate characteristics to the fermentative ability of rumen microbiota while environmental conditions are maintained at an optimal range is a key factor for maximising the efficiency of energy utilisation, and the risk of physiological impairment is also reduced. The characteristics of the specific rumen microbial community promoted by a certain diet also affect substrate utilisation [7], as the activity of the bacterial species able to fermenting starch or fibrous polysaccharides depends on environmental characteristics [8,9]. 

The comparison of these energy sources and their effects in the rumen under in vivo conditions is laborious and expensive, and is often biased by the feeding pattern and hardly controlled fermentation conditions [10]. On the other hand, in vitro studies are cheaper and faster and allow for a good insight into rumen fermentation processes [11]. However, most of these in vitro methods are designed to mimic the environment promoted by high forage diets, including the use of inoculum from forage-fed animals [12], and it is not easy to adapt the main physiological conditions, such as pH and rate of passage to conditions promoted by high-concentrate diets [13]. Amanzougarene and Fondevila [14] succeeded in maintaining a low incubation pH in an in vitro closed-batch system by reducing the bicarbonate concentration in the incubation solution, allowing to compare the fermentation of different carbohydrate sources under conditions simulating high-concentrate feeding [15,16]. However, this is not the real physiological situation in vivo, as pH changes across a wide range throughout the day [17] and, besides, rumen outflow rate cannot be assessed in this system. In this regard, the semicontinuous incubation system [18] modified by Prates et al. [19], applying the procedure proposed by Amanzougarene and Fondevila [14] for controlling incubation pH, appears to be a useful tool to mimic the rumen pH pattern and liquid outflow rates under in vitro conditions.

Therefore, in a semicontinuous in vitro incubation system, we compared the acidification potential and the rumen microbial fermentation pattern of several carbohydrate sources of variable composition when a different rumen environment is promoted by either high-forage or high-concentrate diets, aiming to minimise, where possible, the risk of acidosis during feeding transition from a fibrous to a high-concentrate diet. 

## 2. Materials and Methods 

### 2.1. Substrates and Inocula

Six carbohydrate sources were chosen as substrates: three cereal grains (barley var. Gustav (B), maize var. Dekalb 6667Y (M), and a brown sorghum of unknown variety (S)) and three by-product feeds (sugarbeet pulp (BP), citrus pulp (CP) and wheat bran (WB)). All substrates were ground in a hammer mill (Retsch Gmbh/SK1/417449, Haan, Germany) through a 1 mm sieve. The chemical compositions of the substrates are given in Table 1.

Rumen fluid was obtained from six lambs housed in the facilities of the Servicio de Apoyo a la Experimentación Animal of the Universidad de Zaragoza. The animal care and procedures for extraction of rumen inoculum were approved by the Ethics Committee for Animal Experimentation. The care and management of animals agreed with the Spanish Policy for Animal Protection RD 53/2013, which complies with EU Directive 2010/63 on the protection of animals used for experimental and other scientific purposes. The lambs were weaned at 49 ± 8 days (average weight 13.6 ± 0.78 kg) and, thereafter, three lambs (1, 2 and 3) were fed ad libitum with a concentrate mixture (composed of barley, maize, wheat, and soybean meal) and barley straw (88:12 concentrate to straw ratio) for 35 days, and then slaughtered (average weight 20.6 ± 1.85 kg) to obtain concentrated inoculum (CI). The other three lambs (4, 5 and 6) were fed ad libitum with alfalfa hay and slaughtered after 45 days (average weight 16.5 ± 0.33 kg) to obtain forage inoculum (FI). The rumen contents of each animal were individually filtered through a cheesecloth and dispensed in 16 mL aliquots into 110 × 16 mm tubes, which were immediately frozen in liquid nitrogen and preserved at −80 °C until use [19]. Immediately before incubation, the rumen inoculum was thawed in a water bath at 39 °C (about 2 min).

### 2.2. Experimental Conditions

The in vitro semicontinuous system of Fondevila and Pérez-Espés [18], modified by Prates et al. [19], was used. The substrate samples (800 mg) were dispensed into 4 × 4 cm nylon bags (45 µm pore size) that were sealed and introduced into duplicated bottles (116 mL total volume). The bottles were filled under CO_2_ flux with 80 ml of incubation solution, including 16 mL (0.20 of total volume) thawed rumen inoculum, without resazurin and microminerals [20], and were incubated in a water bath at 39 °C for 24 h in three incubation series, each corresponding to a different donor animal, for each type of inoculum. The buffer solution was modified to include 0.006 M bicarbonate ion in order to get a poorly buffered medium [14].

The pressure produced on each bottle was measured every 2 (from 0 to 12 h incubation) or 4 h (from 12 to 24 h) with a HD8804 manometer provided with a TP804 pressure gauge (DELTA OHM, Caselle di Selvazzano, Italy). The readings, corrected for the atmospheric pressure, were converted to volume (ml) using a pre-established linear regression recorded in this type of bottle (*n* = 48, R^2^ = 0.993), and were expressed per unit of incubated organic matter (OM). Along the incubation, an aliquot volume of the medium was extracted immediately after each gas measurement and replaced anaerobically by the same volume of incubation solution (without microbial inoculum) to simulate an approximate liquid turnover rate of 0.08/h. In order to simulate daily rumen pH fluctuations, from 0 to 6 h, the incubation solution was poorly buffered, as explained above, to allow the incubation pH to drop as fermentation proceeded, whereas, from 8 h onwards, the replacing incubation solution was made up with 0.058 M bicarbonate ion to allow the pH to increase to around 6.5.

The incubation pH was recorded on every extraction. In addition, the medium was sampled at 6 and 10 h for determination of volatile fatty acids concentration (VFA; 2 mL on a 0.5 mL solution of 0.5 M phosphoric acid with 1 mg 4-methyl-valeric acid as the internal standard) and at 6 h for the determination of the lactic acid concentration (2 mL). The samples were stored at -20 °C until analysis. Moreover, another sample (6 mL) was also taken at 8 h and immediately frozen (−80 °C) for the determination of microbial biodiversity by terminal restriction fragment length polymorphism (tRFLP). At the end of incubation, the substrate bags were removed from the bottles, rinsed and dried at 60 °C for 48 h for the determination of dry matter disappearance (DMd).

### 2.3. Chemical and Microbiological Analyses

The dry matter (DM) and OM content in the substrates and the incubation residues were analysed following the AOAC [21] procedures (methods ref. 934.01 and 942.05). The substrates were also analysed for crude protein (CP) and ether extract (EE) (ref. 976.05 and 2003.05) [21], and their concentration of neutral detergent fibre (aNDFom) was analysed as described by Mertens [22] in an Ankom 200 Fibre Analyser (Ankom Technology, New York, NY, USA), using α–amylase and sodium sulphite, with results being expressed exclusive of residual ashes. The acid detergent fibre (ADF) (ref. 973.18) and acid detergent lignin (ADL) were determined as described by AOAC [21] and Robertson and Van Soest [23], respectively. Neutral detergent soluble fibre (NDSF) was estimated following Hall et al. [24], discounting the aNDFom and the ethanol insoluble EE, CP and starch fractions from the insoluble OM. The total starch content in B, M, S and WB substrates was determined enzymatically from samples ground to 0.5 mm using a commercial kit (Total Starch Assay Kit K-TSTA 07/11, Megazyme, Bray, Ireland). The total phenolic (TP) content in S was analysed following the colourimetric method of Makkar et al. [25] using the Folin–Ciocalteau reagent and with tannic acid (MERCK Chemicals, Madrid, Spain) as the reference standard. The total tannins (TT) were estimated as the difference between TP before and after treatment with polyvinyl polypyrrolidone. 

The frozen samples of medium incubation were thawed and centrifuged at 13,000 *g* for 15 minutes at 4 °C in order to analyse their lactic acid and VFA. The VFA were determined by gas chromatography on an Agilent 6890 apparatus equipped with a flame detector and a capillary column (HP-FFAP Polyethylene glycol TPA, 30 m × 530 µm id). The lactic acid concentration was determined by the colourimetric method proposed by Barker and Summerson [26]. For the microbial diversity analysis, frozen microbial samples were freeze-dried, thoroughly mixed and disrupted (Mini-Bead Beater, Biospec Products, Bartlesville, OK, USA). The DNA was extracted using the Qiagen QIAmp DNA Stool Mini Kit (Qiagen Ltd., West Sussex, UK) following the manufacturer’s recommendations, except that the samples were initially heated at 95 ºC for 5 min to maximise the lysis of the bacterial cells. The concentration of extracted DNA was tested in a Nanodrop ND-1000 (Nano-Drop Technologies, Inc., Wilmington, DE, USA). PCR was performed using a 16S rRNA bacteria-specific primer (cyanine-labelled forward 27F, 5′-AGA GTT TGA TCC TGG CTCAG-3′ and unlabelled reverse 1389R, 5′-AGG GGG GGT GTG TAG AAG-3′; [27]) using a DNAEngine^®^ Gradient Cycler (Bio-Rad, Spain). The PCR product was purified using a Purelink PCR purification kit (ref. K3100-01; Invitrogen) and diluted to 10 µL. The DNA concentration of each amplified and purified sample was obtained by spectrophotometry (Nanodrop^®^ ND-1000 spectrophotometer) to enable a standardised quantity of 50 ng DNA to be used per restriction enzyme digest reaction. The digestion of samples was carried out using HhaI, HaeIII and MspI (Promega, Spain), following the manufacturer recommendations, except for HhaI, where the recommended addition of bovine serum albumin was omitted. The restriction digests were purified by ethanol precipitation [28] in 35 μL sample loading solution buffer, including a 600 bp size standard (Beckman Coulter Inc., Fullerton, CA, USA) before being applied to a 3500 × L Genetic Analyzer (Applied Biosystems). Once the size and height of every peak was obtained, 1% of the second highest peak was used as the criteria for the lower threshold for peaks, to detect and eliminate smaller, broader peaks that would not be indicative of single true OTUs. 

### 2.4. Calculations And Statistical Analyses

The tRFLP results were analysed from a matrix generated for each data list obtained, and results were presented in the form of relative abundance. The three matrices resulting from each series and enzyme were concatenated and analysed with R statistical software (https://cran.r-project.org/bin/windows/base/, version 3.5.0, R Foundation for Statistical Computing, Vienna, Austria). FactoMineR, Factoextra, MixOmics, Vegan, MASS, and Ggplot2 packages were used to carry out the analysis of hierarchical classification on the principal components for obtaining the cluster dendrogram. 

The results were analysed statistically by ANOVA using the Statistix 10 package [29]. On each sampling time, the effect of the incubation series (equivalent to the donor animal; interaction inoculum x incubation series, random effect), the type of inoculum, the type of substrate, and the interaction of both factors on pH, gas production, total VFA and lactic acid concentration, and VFA profile were studied as factors. The treatment differences among the means with *p* < 0.05 and 0.05 < *p* < 0.10 were accepted as representing statistically significant differences and a trend to the differences, respectively. When significant, the differences were contrasted by the Tukey *t*-test. Simple and multiple linear regressions were established to study the relationships among the different parameters studied.

## 3. Results 

### 3.1. Pattern of Incubation pH

The mean inoculum pH at the start of the incubation series was 6.45 ± 0.15 and 6.87 ± 0.02 for CI and FI, respectively (*n* = 3). The average minimum pH was recorded at 6 h incubation (5.96) for CI, and at 8 h (6.22) for FI. Thereafter, the pH increased to reach its maximum (6.64 for both inocula) at 24 and 20 h for CI and FI. The pH differences in the incubation medium among inocula (*p* < 0.05) were ± 0.3 units from 2 to 6 h, decreasing gradually to ± 0.1 at 12 h. A significant interaction inoculum × substrate (*p* < 0.05) observed on pH at 4, 8, 10, 12, 16 and 20 h and a tendency (*p* = 0.052) at 2 h incubation indicates the different behaviour of the substrates depending on the inoculum. Therefore, a comparison of the pH pattern among the incubated substrates is presented in Figure 1 separately for each inoculum. With CI (Figure 1a), the lowest incubation pH from 2 to 12 h was recorded with CP (*p* < 0.05), reaching its minimum at 6 h (5.60), although recovered thereafter to 6.63 at 24 h incubation. In ascending order, WB and B reached their minimum pH at 8 h (5.89 and 5.97, respectively), whereas BP, M, and S maintained a higher medium pH from 4 to 8 h (*p* < 0.05). The differences among M, S, BP, and B disappeared from 10 to 16 h (*p* > 0.05), and no differences were detected among the substrates (*p* > 0.05) at the end of incubation. When the substrates were incubated using FI (Figure 1b), CP recorded the lowest pH from 4 to 10 h incubation (*P* < 0.05), and its minimum value was 5.90, whereas S, M and BP maintained the highest medium pH during this period (6.30 to 6.46), and B and WB were grouped at intermediate values (*p* < 0.05). At 16 and 20 h incubation only, B recorded a lower value (6.44 and 6.57; *p* < 0.05) and, again, no differences were detected among the substrates at the end of incubation (*p* > 0.05). 

### 3.2. Pattern of in Vitro Gas Production

The volume of gas produced with the CI inoculum was higher than that obtained with FI at all incubation times (*p* < 0.05). Because of the interaction inoculum x substrate at 4 h and from 8 to 24 h (*p* < 0.05), for an easier understanding, the gas production is presented separately for CI and FI (Figure 2). The major difference among substrate fermentative behaviour between the inocula is manifested in the magnitude of differences among them. Thus, with CI (Figure 2a), CP recorded the highest gas volume from 4 h onwards, at 12 h being on average 0.42 times higher than the other substrates, while also recorded differences at 2 h with BP and S (*p* < 0.05). The gas volume with WB was higher than BP and S from 4 h onwards, and higher than M from 6 h and B from 8 to 20 h (*p* < 0.05). Differences were also recorded between B and S from 8 to 16 h and at 24 h (*p* < 0.05). A similar pattern was observed with FI (Figure 2b), but the magnitude of differences was lower. Thus, CP was higher than B, M, BP and S from 6 to 24 h (*p* < 0.05), with differences at 12 h reaching 0.59 of their average, but did not differ from WB, which was higher than BP and S in that period and also higher than M from 6 to 10 h (*p* < 0.05). Differences between B and M respect to S were also detected from 16 and 20 h onwards, respectively (*p* < 0.05). 

### 3.3. Dry matter Disappearance (DMd)

Inoculum differences in DMd after 24 h of incubation were not detected, although CI was numerically higher than FI (proportions of 0.382 *vs*. 0.339 from the substrate weight; *p* > 0.05). The substrates ranked according to the proportion of DMd as follows: CP, 0.502 > B, 0.449 > WB, 0.360, M, 0.343 > BP, 0.265, S, 0.243 (*p* < 0.001; SEM = 0.0120). The interaction inoculum x substrate was not significant (*p* = 0.21), indicating that the substrates behaved similarly with both inocula.

### 3.4. Volatile Fatty Acids and Lactic Acid Production

Table 2 and Table 3 show that CI promoted a higher (*p* < 0.05) concentration of total VFA than FI at both 6 (23.2 *vs.* 9.8 mM) and 10 h (22.2 *vs.* 9.3 mM). The molar proportions of acetate, propionate, and butyrate did not manifest the differences between inocula (*p* > 0.05), whereas with CI, valerate was higher and branched-chain volatile fatty acids (BCVFA, sum of isobutyrate and isovalerate) lower than with FI at both incubation times (*p* < 0.05). 

Among the substrates, at 6 h (Table 2) CP recorded a higher total VFA concentration than BP, M and S (average values of 19.3, 15.6, 14.5 and 15.0 mM, respectively), whereas WB (15.9 mM) was also higher than M and S (*p* < 0.05). Differences in the molar VFA profile were only recorded for BCVFA, with the highest proportions in S and M and the lowest with CP (*p* < 0.05); however, the interaction inoculum x substrate (*p* < 0.001) indicates that differences in BCVFA proportion were only observed with FI. Regarding the concentration of lactic acid at 6 h among the substrates, CP recorded the highest concentration and BP and S the lowest (8.7 *vs.* 1.3 and 1.4 mM; *p* < 0.05). Similar trends were observed at 10 h in total VFA concentration (Table 3), with CP rendering a higher concentration than BP, M and S, but tending to be significant only with CI interaction inoculum x substrate, (*p* = 0.058). The interaction inoculum x substrate in the proportion of propionate (*p* = 0.017) indicates that values recorded with WB and CP were higher than those with BP and S, but differences were only manifested with FI, whereas no differences (*p* > 0.05) among the substrates were recorded on acetate, butyrate and valerate proportions. The highest proportion of BCVFA was promoted by S and the lowest by CP (*p* < 0.05). 

### 3.5. Bacterial Biodiversity

Bacterial biodiversity after 8 h of incubation was markedly affected by the source of rumen inoculum. Thus, the substrates incubated with rumen inoculum from lambs fed the high-concentrate diet clustered together, except for WB in the first incubation run, as well as substrates that were incubated with FI (Figure 3). Bacterial biodiversity was also markedly affected by the incubation series—that is, the donor animal—for both inocula.

## 4. Discussion

Conventional in vitro closed batch systems are adapted for the study of microbial fermentation under conditions mimicking high forage diets, which is not applicable to evaluate diets given in intensive ruminant-fattening systems. An in vitro semicontinuous incubation system [18,19], adapted to control of the pH by modifying the bicarbonate ion concentration [14] allows us to approach the ruminal fermentation pattern of the different carbohydrate sources to the rumen physiological conditions that occur in intensive feeding systems, either during a transition process to high-concentrate diets (i.e., when rumen conditions are still modulated by a forage diet) or when animals are adapted to such feeding conditions (as promoted by a concentrate diet). The pH pattern obtained along the in vitro incubation with CI and FI, reaching a minimum value at 6–8 h after substrate availability and then progressively increasing to final pH values of around 6.4–6.5, fitted well with the circadian evolution of rumen pH observed with practical forage or concentrate feeding of ruminants [30]. Thus, we can assume it allows the different substrates to express their acidification potential at the time that their fermentation is compared under more realistic conditions. 

### 4.1. Effect of the Inoculum Source on the in Vitro Fermentation Pattern

The source of rumen fluid has an important role in the pattern of in vitro fermentation [15,31,32], with an inoculum promoted by a concentrate diet having a higher fermentative potential than another from a forage diet. In our experiment, the lower buffering of the incubation medium during the first 6–8 hours allowed for a clear expression of the acidification potential of the incubated substrates, which was expressed at a higher extent with CI than FI (average pH along the 24 h incubation period ranging from 6.45 to 5.96 *vs.* 6.87 to 6.22) as pH dropped to values close to those considered as a threshold for microbial activity [33], whereas a higher pH was maintained with FI. Despite this, substrates incubated with CI rendered almost two-fold gas volume more than with FI, irrespective of the chemical nature (starch- or fibre-rich) of those substrates. Despite the more pronounced drop of pH with CI, the incubation environment promoted by a concentrate diet given to the donor animals was more favourable for fermentation of non-fibrous carbohydrates than that induced by a fibrous diet [20,34], probably because of the lack of adaptation of microbiota to ferment starch and sugar substrates with a forage inoculum [15,35] and the inherent buffering capacity of forage legumes such as alfalfa. However, assuming that a part of the gas produced comes from the activity of bicarbonate ion in the buffering of fermentation acids produced, such differences in gas production could be partly associated with the lower pH promoted by CI inoculum, although the contribution of this indirect gas is hard to quantify [14]. In the case of the byproducts, characterised by their richness in rapidly fermentable fibre, microbiota might easily counterbalance the lack of adaptation for their degradation [36,37]. In contrast, the low pH occurring during the initial part of incubation may affect, at a higher extent, the activity of the bacterial species adapted to fibre degradation, causing a lower magnitude of fermentation of structural polysaccharides like cellulose and hemicelluloses [12,38,39]. 

Contrary to what might be expected, the results of the gas production were not supported by those of DMd. This parameter was especially low compared to the extent of the rumen degradation of starch-rich sources (around 0.70–0.80 [40]) or fibrous sources (ranging from 0.40 to 0.70 [41]). This is difficult to explain, but we have also observed this low response in previous in vitro experiments [42], partly associated with a low pH [13]. Calsamiglia et al. [43] justified similar results by the differences between rumen and in vitro microbial ecosystems, partly because the dilution of inoculum in the latter reduces the extent of the degradation. In contrast to DMd, the concentration of total VFA followed a similar trend than that of gas production, being higher for CI at both sampling times, as has been observed by others [15,32,43]. Calsamiglia et al. [43] did not observe any inoculum effect on the acetate and butyrate proportions, and propionate proportion was higher with the concentrate inoculum, as observed in our study at 10 h incubation. However, differences in the proportion of BCVFA, which resulted from fermentation of protein and branched-chain amino acids [44] were higher with FI between 6 and 10 h, probably because of the fermentation of protein from the alfalfa hay fed to the donor lambs. The effect of the inoculum source was also observed in microbial diversity, suppporting the recent findings reported by Tapio et al. [45] and Nagata et al. [46] who showed the difference in the rumen microbial population when bulls were fed with forage or concentrate diets.

### 4.2. Effect of Different Substrates on the In Vitro Fermentation Pattern 

Despite the marked differences in the magnitude of fermentation between CI and FI, the fermentation pattern among the substrates was almost the same between both inocula. The results of the measured parameters showed a strong correlation between gas production and the other parameters (pH, VFA and lactic acid concentrations) at 6 h (*n* = 36; adjusted R^2^ = 0.90; *p* < 0.001). Similarly, at 10 h incubation, the volume of gas produced was strongly correlated with incubation pH and VFA (*n* = 36; adjusted R^2^= 0.84; *p* < 0.001). These results confirm that the gas production and, equally, the concentration of total VFA and lactic acid, are the main factors indicating the acidification potential of the incubated substrates [31,47,48]. Citrus pulp had a higher acidification capacity than the other substrates, which is associated with a higher magnitude of fermentation that is manifested in high gas production, as well as VFA and lactic acid concentration. Despite the high concentration of lactic acid with CP at 6 h (Table 2), it did not achieve the range considered as a risk of acidosis in vivo [30] and, in fact, did not promote the values of incubation pH below 5.5 that are considered as a threshold for the onset of subacute acidosis [49]. These results were in agreement with those found by Amanzougarene et al. [42] in a batch culture with a minimum buffer concentration, and could be associated with its richness in soluble sugars [50,51], estimated as 0.24 g/kg DM (Table 1), which are fermented at a very fast rate. Although CP also has a high proportion of soluble fibre (0.42, Table 1), this response cannot be directly associated with the fast fermentation of pectin [37,52], since BP includes a similar NDSF proportion (0.46) and it was fermented at a slower rate and magnitude. In fact, Strobel and Russel [53] reported that at a pH of 6.00, the extent of pectin fermentation was reduced with respect to a higher pH. The lower fermentation rate of BP and, thus, its lower acidification potential can also be related to its high NDF content, which does not ensure its maximum fermentation in the 24 h incubation period [54]. Considering the aforementioned characteristics of BP composition, mainly its high NDF and NDSF proportions as well as its low sugar content, its lower concentration of lactic acid produced with respect to the other incubated substrates could be expected. Others [52,53] have also stated that the yield of lactic acid production from pectins fermentation is very low.

The extent of fermentation of WB and B was lower than that of CP, but higher than those of the remaining substrates, probably linked to the high proportion of rapidly fermentable starch in these substrates, compared with those of M and S. Nocek and Tamminga, [55] indicated that 0.80 to 0.90 of barley or wheat starch is digested in the rumen, compared to only 0.55 to 0.70 of that of corn and sorghum. In addition, WB have a considerable amount of NDSF and highly fermentable NDF (Table 1). The structure of the starch endosperm of maize and sorghum, together with their different proportions of amylose [4], as well as the protein matrix in the endosperm in these cereal species [56] and the presence of phenolic compounds in the brown sorghum [16] explain why the fermentation of starch of barley and wheat bran by ruminal bacteria was higher [4,57,58]. Consequently, the differences in the starch characteristics and fermentation rate promote the response in a medium pH [15]. When incubating several grains in a well buffered medium, Lanzas et al. [59] observed a higher fractional rate of 48 h gas production with barley than maize and sorghum varieties (on average, 0.24, 0.15 and 0.06/h). Opatpatanakit et al. [60], modified the incubation pH similarly to the present work, and also observed the highest gas production with barley, intermediate with maize and lowest with sorghum (on average, 222, 138 and 104 mL/g DM, respectively), under pH values at 7 h incubation ranging from 5.7 to 6.1 for barley, 6.5 to 6.9 for maize and 6.5 to 6.8 for sorghum.

From our findings, it can be indicated that citrus pulp and, to a lower extent, wheat bran had an acidic capacity of an even higher magnitude than cereal sources, including barley. Despite the differences on the magnitude and extent of fermentation between the different incubated substrates, the results of microbial diversity with both inocula showed the major effect of the donor animal on this parameter, partly because of aspects related to in vitro methodology, such as the short period of incubation. These results were in accordance with those reported by Taxis et al. [61] and Söllinger et al. [62], explaining the differences in microbial diversity from one animal to another. However, within each series (donor animal), our results did not demonstrate differences between the substrates. 

## 5. Conclusions

Under the fermentation conditions of high-concentrate feeding, some sources of highly fermentable fibre, such as citrus pulp and, to a lower extent, wheat bran may create a more acidic environment than cereals. Among these, barley promotes a lower pH than maize or sorghum, as this grain is associated with a higher rate and extent of fermentation. The rumen environment promoted by high forage/fibre diets is not adapted for non-fibrous carbohydrates, and fermentation of soluble fibre is not differentially enhanced, producing a lower extent of substrate fermentation than concentrate diets. Therefore, the choosing of ingredients is important when ruminants are changed from a forage to a high-concentrate diet, although this cannot be inferred from this study. In any case, in this experiment, acidification levels did not reach those that may change the fermentation pattern. Care must be taken in substrate comparison in terms of gas production, since the buffering of the medium under low pH conditions may overestimate the fermentation differences by increasing the indirect gas production. The in vitro semicontinuous system, adapted to a variable medium pH, has proven to be useful for the study of rumen microbial fermentation under intensive feeding conditions.

## Figures and Tables

**Figure 1 animals-10-00261-f001:**
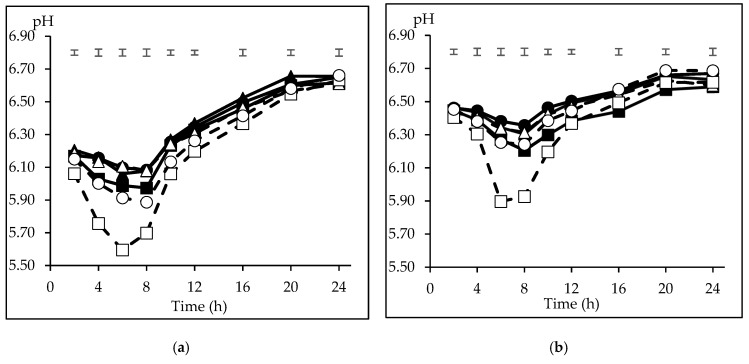
Pattern of medium pH of carbohydrate substrates (barley (B) ■, maize (M) ▲, sorghum (S) ●; solid lines, citrus pulp (CP) ☐, sugar beet pulp (BP) △, wheat bran (WB) ○; dashed lines) incubated with inoculum from concentrate (CI, Figure 1a) or forage (FI, Figure 1b) diets. The initial pH was 6.45 (Figure 1a) and 6.87 (Figure 1b). The upper bars show the standard error of the means (*n* = 3).

**Figure 2 animals-10-00261-f002:**
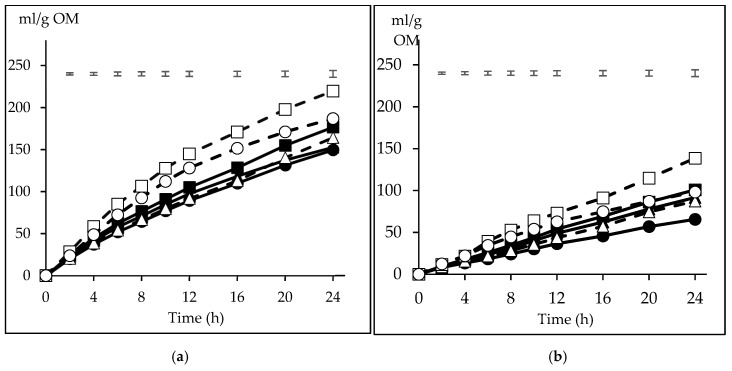
Pattern of gas production from the carbohydrate substrates (barley (B) ■, maize (M) ▲, sorghum (S) ●; solid lines, citrus pulp (CP) ☐, sugar beet pulp (BP) △, wheat bran (WB) ○; dashed lines) incubated with inoculum from concentrate (Figure 2a) or forage (Figure 2b) diets. The upper bars show the standard error of the means (*n* = 3).

**Figure 3 animals-10-00261-f003:**
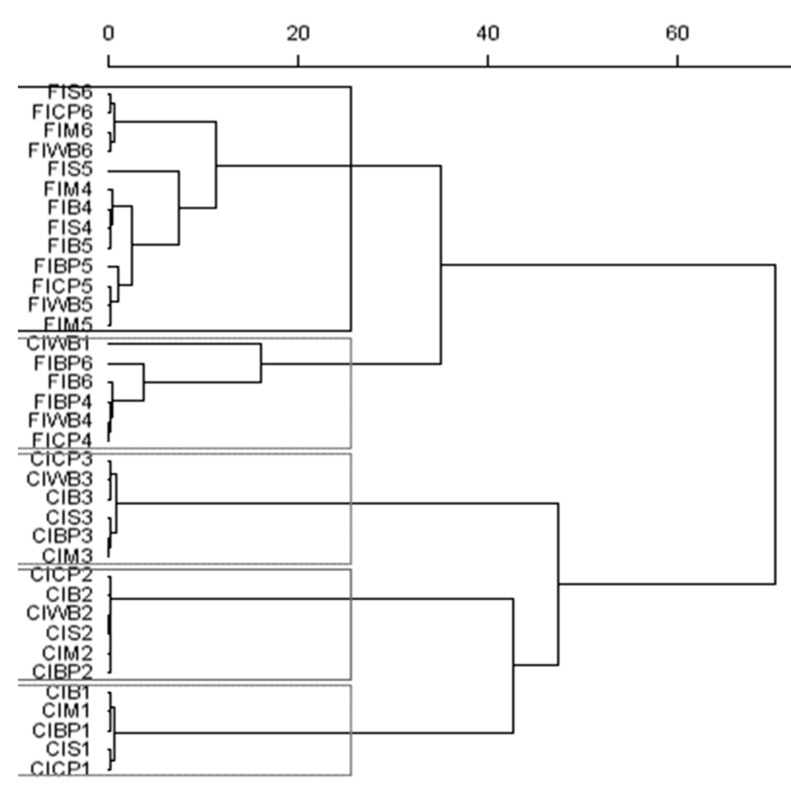
Dendrogram of bacteria diversity from terminal restriction fragment length polymorphism (tRFLP) data generated by enzyme digestion (HhaI, MspI, and HaeIII) for the carbohydrate substrates (B, M, S, BP, CP, and WB) incubated for 8 h with inoculum from concentrate (CI) or forage (FI) diets. The scale bar shows the Euclidean distances, “Ward method”.

**Table 1 animals-10-00261-t001:** Chemical composition (g/kg DM) of feeds used as incubation substrates.

Component	B	M	S	BP	CP	WB
OM	978	986	979	953	940	944
CP	105	75	113	107	59	161
EE	24	34	11	5	14	31
Starch	672	706	647	-	-	245
aNDFom	173	91	97	437	207	499
ADF	56	25	60	272	192	145
ADL	18	2	5	75	21	37
NDSF	4	77	110	457	423	155
Sugars	1.6	13	1.3	9	243	31
TP	-	-	2.6	-	-	-
TT	-	-	1.3	-	-	-

Barley (B); maize (M); sorghum (S); sugar beet pulp (BP); citrus pulp (CP); wheat bran (WB). Dry matter (DM); organic matter (OM); crude protein (CP); ether extract (EE); neutral detergent fibre (aND­Fom); acid detergent fibre (ADF); acid detergent lignin (ADL); neutral detergent soluble fibre (NDSF). Total phenolics (TP); total tannins (TT).

**Table 2 animals-10-00261-t002:** Average of total volatile fatty acids concentration (VFA, mM) and molar VFA proportions (mmol/mmol), together with lactate concentration (mM) recorded at 6 h of the different carbohydrate sources incubated as substrates with concentrate (CI) or forage (FI) inoculum.

Substrates	VFA	Acetate	Propionate	Butyrate	Valerate	BCVFA	Lactic acid
with CI							
B	22.34 ^ab^	0.570	0.240	0.153	0.022	0.014	3.83 ^b^
M	20.18 ^bc^	0.578	0.235	0.149	0.023	0.016	2.35 ^c^
S	21.72 ^abc^	0.588	0.245	0.131	0.022	0.015	1.93 ^c^
BP	21.66 ^abc^	0.593	0.235	0.135	0.021	0.016	0.90 ^c^
CP	26.55 ^a^	0.595	0.238	0.135	0.020	0.012	8.70 ^a^
WB	26.94 ^a^	0.590	0.245	0.132	0.020	0.013	2.95 ^c^
With FI							
B	10.31 ^xyz^	0.633	0.226	0.102	0.009	0.030 ^y^	3.05 ^y^
M	8.83 ^yz^	0.632	0.229	0.095	0.010	0.034 ^xy^	2.69 ^y^
S	8.31 ^z^	0.642	0.216	0.094	0.009	0.039 ^x^	0.84 ^y^
BP	9.52 ^yx^	0.665	0.207	0.088	0.008	0.032 ^y^	1.64 ^y^
CP	12.12 ^yz^	0.686	0.209	0.075	0.008	0.023 ^z^	8.67 ^x^
WB	9.83 ^yz^	0.653	0.225	0.083	0.009	0.030 ^y^	2.97 ^y^
SEM	1.065	0.0156	0.0076	0.0085	0.0010	0.0009	0.588
*p*-Value							
Inoculum	0.002	0.077	NS	NS	0.005	<0.001	NS
Substrate	<0.001	NS	NS	NS	NS	<0.001	<0.001
Inoc. × Subs.	NS	NS	NS	NS	NS	<0.001	NS

Means within a column with different superscripts for CI (^a,b,c^) or FI (^x,y,z^) differ (*p* < 0.05). Standard error of the means (SEM). Branched-chain volatile fatty acids (BCVFA) (sum of isobutyrate + isovalerate). NS: *p* > 0.10.

**Table 3 animals-10-00261-t003:** Average of total volatile fatty acids concentration (VFA, mM) and molar VFA proportions (mmoL/mmoL), recorded at 10 h of the different carbohydrate sources incubated as substrates with concentrate (CI) or forage (FI) inoculum.

Substrates	VFA	Acetate	Propionate	Butyrate	Valerate	BCVFA
With CI						
B	21.16 ^b^	0.561	0.225	0.172	0.028	0.014
M	19.42 ^b^	0.548	0.227	0.178	0.031	0.017
S	19.87 ^b^	0.557	0.249	0.152	0.026	0.017
BP	20.25 ^b^	0.604	0.229	0.130	0.022	0.016
CP	27.32 ^a^	0.553	0.240	0.164	0.031	0.012
WB	25.10 ^ab^	0.537	0.261	0.158	0.029	0.015
With FI						
B	10.46	0.597	0.256 ^xy^	0.117	0.009	0.022
M	8.51	0.620	0.236 ^xy^	0.109	0.009	0.026
S	8.48	0.620	0.222 ^y^	0.120	0.009	0.029
BP	8.34	0.611	0.227 ^y^	0.126	0.009	0.027
CP	10.47	0.579	0.267 ^x^	0.124	0.010	0.019
WB	9.76	0.575	0.274 ^x^	0.117	0.010	0.025
SEM	1.137	0.0233	0.0078	0.0169	0.0019	0.0024
*p*-Value						
Inoculum	0.011	NS	NS	NS	0.008	0.008
Substrate	0.001	NS	<0.001	NS	NS	0.046
Inoc. × Subs.	0.058	NS	0.017	NS	NS	NS

Means within a column with different superscripts for CI (^a,b,c^) or FI (^x,y,z^) differ (*p* < 0.05). SEM: standard error of the means. Branched-chain volatile fatty acids (BCVFA) (sum of isobutyrate + isovalerate).

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
