# Peer review of "Fermentation Pattern of Several Carbohydrate Sources Incubated in An in Vitro Semicontinuous System with Inocula From Ruminants Given Either Forage or Concentrate-Based Diets"

_animals, 2020, doi:10.3390/ani10020261_

Round 1
Reviewer 1 Report
This study deals with the evaluation in an adapted in vitro system of the fermentation pattern of various carbohydrates sources, usually used in formulation, with the aim of analyse the acidification potential of the substrates with two different inocula (adapted or not to high concentrate diets). The study is very interesting and fits with the scope of the journal. The implication of the obtained results could be of interest for nutritionists when using these feedstuffs in the rations for feedlot or fattening of ruminants. The introduction in general terms is sound and well written. However, Authors did not achieve to highlight the implications and main objective of the study. In vitro trial design is correct and well described in the material and methods section: Maybe it would have been also interesting to test in vitro the fermentation kinetics of the different substrates. Results should be presented in a different way because it is a factorial design. Discussion is well written and in general succeeded in discussing scientifically the observed findings in a biologically integrated fashion, both within the study as well as relative to results of other scientists. Overall, the paper has certain novelty and is well written, but I recommend the Authors to consider the following remarks to improve the quality of the manuscript.
Specific comments
Introduction
L 47-5: This paragraph considers revision, too long and confusing.
L52 consider replacing “at the time” for “while”
L60-61: Consider this redaction: “On the other hand, in vitro techniques are cheaper and faster and allow for a good insight into rumen fermentation processes”
L62: promoted
L68-70: You used a “semi-continuous” system where a rumen outflow rate can be mimicked, didn’t you? (L74-77). I think you can rewrite these two paragraphs together.
I only missed in this section a clear description of the novelty or major implications of the study, and a clear statement of the objective of the study (to test the acidification potential, or minimise risk of acidosis during transition, to test the in vitro semicontinuous system..?)
Materials and methods.
L91-92: In CI please describe what is the forage:concentrate ratio and the proportions of the ingredients of the concentrate mixture
L94: for FI why was alfalfa the forage used? If you want to mimic the transition from pasture to a concentrate diet, why you don’t use grass hay for example..?
L102: please indicate the volume of the bottles
L112: please indicate the volume of the aliquot
L115: units for turnover rate
L119: remove “Besides”
L124: Before rinse the bags it is normal to fridge them in order to help bacteria dis-attachment from the bags.
L144: please indicate the detector used in the chromatograph
L166: TRFLP: this acronym is not described previously
L169: you have to include a citation for every package used in the analysis
L172: please indicate the model. What is the experimental unit?
L173: I don’t understand the use of the term block here
L175: this is a factorial analysis, please indicate, indicate also the random effect (if exits)
L182-189: comment first the interaction
L184: incubation medium and not inoculum
L186: 4, ….,20 h and a tendency at 2h of incubation indicates….
L200: indicate units for volume of gas produced
L204: In the figures we cannot appreciate the magnitude of differences between substrates being different for the two inoculum used. The same in L208. Maybe you can indicate the differences in %
L212-217: I have some doubts with this section
-Please indicate the units for DMd
-DMd values are surprisingly low for this kind of substrates. It makes me doubting about the validity of these results.
-L216: was not significant
L 218-235: Change BCFA for BCVFA (branched-chain volatile fatty acids) here and elsewhere
There are some differences both at 6 and 10 H that only can be seen in FI inoculum (L226-227 and 233). We cannot check this fact in the tables. I think that the presentation of the results in the Tables is not correct. You have to present them in order the person who read the paper would be able to see the interactions. Results have to be presented with one effect within other. In factorial designs, for example a 3×2 factorial should generally be presented in tables as six data columns with either the second factor means listed within the first factor, or vice versa. In general, main effect means should not be pooled since it is always possible for readers to calculate main effect means from the individual means within factors, but not possible to calculate the individual means within factors from pooled means. There should also be a column to express an S.E. or S.E.M. and, in the case of the 3×2 factorial, columns to show the P values for each main effect and the interaction. In cases where the interaction is deemed statistically significant, then a secondary statistical test should be used to separate the efficacy of factor 2 within factor 1 (or vice versa). For more information please read (Some experimental design and statistical criteria for analysis of studies in manuscripts submitted for consideration for publication. Editorial. Animal Feed Science and Technology 129 (2006) 1–11).
L236-241: different spaced was used here
L240: for FI is more difficult to see the clustering of the series.
Figures 1 and 2: include axes titles. Maize is not well seen in the Tables. In Figure 1 include “medium pH” instead of “pH”
Table 1. describe B, M, … in the footnotes. Please remove decimals from ADL
Table 2 and 3: please change the tables as suggested in a previous comment. You should take into account the factorial design. You have not to define C2, C3 and C4 in the footnotes, remove please. Please use BCVFA instead of BCFA. In Table 3 in the title remove please “lactic acid concentration” since you didn’t measured it at 10h.
Discussion
The ‘Discussion’ section should be fully consistent with the ‘Introduction’ and the aim of the study
L281-285: this is correct but you have to take into account that you increased buffering potential from 8h onwards.
L288: please delete kinetics, you don’t measured kinetics
L290-291: this is a general statement or is for your study?, the CI inoculum have higher fermentative potential with fibrous substrates or forages?
L295: The pH is higher for FI but not more stable because the change was greater (6.45 to 5.96 (8%) vs 6.87 to 6.22 (10%))
L306-309: this is not applicable for CP
L311: is lower fo FI
L310-315: I think you have to discuss more in detail the odd results in disappearance, in other in vitro studies no such low values are obtained, maybe de changes in pH?
L326: remove kinetics
L328-329: If you have performed a correlation analysis please indicate in mat and methods section.
L341: include units for content of soluble sugars
L364: include units
L368-370: your results don’t support this sentence
L373: Why cans microbial diversity showed a major effect of the donor animal due to the dilution of microbial inoculum?
L375-377 is reiterative
A conclusion dealing with the acidification potential of the different substrates in conditions simulating the transition process (FI inoculum) or in conditions when animals are adapted to concentrate diet (CI inocumum) is missing
Author Response
REVIEWER 1: We appreciate your suggestions, that will surely contribute to improve the paper. According to them, we have tried to clarify the objectives of the work done.
Specific comments:
L47-50: this sentence has been checked and redrafted
L52: done
L60-61: done
L62: done
L68-70: sentences have been unified and rewritten accordingly. With changes done at the end of the Introduction section (now in L70-78) the implications of the study and its objective are clarified.
L91-92: the concentrate to straw ratio for CI has been included. The concentrate (additives free) was a commercial product, and as such it did not have specification of ingredients proportion.
L94: alfalfa hay was used because it is the most widely used forage in our region. It was chosen for comparing with a high quality forage that could be given alone (at a similar protein level than CI).
L102: done
L112: included
L115: the turnover rate indicates the proportion of total volume that escapes every hour. As the journal policy does not consider using percentage units (8%/h), this should be the way to present it.
L119: done
L124: considering the dilution of the inoculum (20% of total volume), microbial attachment should be much lower than in situ, so the importance of this factor should be minor. In any case, all bags were equally processed.
L144: the flame detector has been mentioned
L166: meaning of tRFLP abbreviation has been added in actual L129
L169: all these packages are into the R software, so the same internet reference applies for them all
L172: I consider the model is not necessary, as it is explained in the same sentence. A mention to the experimental unit has been included
L 173 and 175: the statistical analysis has been rewritten for clarification, indicating that the incubation run (animal within diet) is a random effect. Mention to the block has been removed.
L82-189: the average (for both inocula) pH pattern is commented first (now in L185-188) to give a general idea of incubation characteristics, and then the interaction is explained (L189 onwards).
L184: the sentence has been rewritten
L186: the sentence has been rewritten accordingly
L200: units for gas production (mL/g OM) are already defined in L112 (now L116). To include it again here should be repetitive
L204: I am not sure that % is accepted by the journal (SI units are recommended). I understand the concern of Reviewer 1 in this regard, but we consider that in terms of fermentation pattern a graph gives a much clearer idea of treatment ranking, even though specific values are not visible. In any case, I have included some mentions to the proportion of CP respect to the other treatments, but it has to be considered that it is difficult to include differences between CP and the other 5 treatments for the 9 times of measurement.
L212-217: DMd values are proportions of initial substrate weight, as it is now indicated. The low magnitude of DMd was already discussed in L311-315, now extended in L322 to 327.
L216: The expression “did not reach significance” has been changed for “was not significant”
L218-235: the abbreviation BCFA has been changed for BCVFA throughout the text. I understand the Reviewer´s concern about tables. We aimed to simplify them considering that the number of significant interactions (3 out of 7 in Table 2 and 3 out of 6 in Table 3), corresponding one of them to BCVFA which is often an artefact due to the low magnitude of values did not deserve such a complicated table, and we preferred including inoculum means because of their major interest. In any case, the suggestion from Reviewer 1 has been accepted, and both tables have been redrafted and text adapted accordingly.
L236-241: this has been corrected
Figures 1 and 2: titles and axes have been corrected following the suggestions. Tables have also been corrected as requested
L281-285: yes, we assume that the acidification capacity of either inocula or substrates can be compared in the first 8 h of incubation, since buffer potential of solution is very low, whereas from 8 h onwards pH must be allowed to recover for the study of the treatments behaviour in real fermentative conditions
L288: the term “kinetics” has been changed for “pattern”
L290-291: it is a general statement, but focused to the type of substrates that are checked here. Obviously, it depends on the nature of the substrate, but in our work CI allowed for a higher fermentation of fibrous substrates (CP, BP and WB) than FI. I cannot ensure what should happen with a high fibre, low quality roughage.
L295: you are right; the sentence has been changed
L306-309: even CP is considered as a fibrous feed, fermentation of NDSF, which is at a higher proportion than NDF, is not so markedly dependent on pH. I have tried to specify it in text
L311: this has been amended
L310-315: apart of what is mentioned, we do not have any solid explanation for this response, but we have repeatedly observed this in previous in vitro experiments. The range of pH may explain differences among results in vitro, but not compared with values in vivo. These comments have been included in text
L326: done
L328-329: done
L341: it is a proportion. Anyway, units are now included
L364: as explained before, turnover rates are given as proportion of total volume renewed per unit of time (h). If percentages should be accepted by the journal, it might be “0.24,0.15 and 0.06%/h)”
L368-370: I disagree; pH pattern in Figs. 1a and 1b show that during the period in which medium was minimally buffered, the drop of pH with CP and, to a lower extent WB (this is now mentioned in text, L380), was lower that with the other substrates, despite the source of inoculum
L373: the mention to inoculum dilution has been removed
L375-377: the sentence has been shortened accordingly
- REVIEWER 1 asks for a conclusion dealing with the behaviour of substrates in a transition from forage to concentrate; however, it must be considered that this has not been studied here (lambs were different between diets, there were no transition from one diet to another). In any case, a mention to this has been now included in Conclusions section.
Reviewer 2 Report
In my opinion the manuscript titled “Fermentation pattern of several carbohydrate sources incubated in an in vitro semicontinuous system with inocula from ruminants given either forage or concentrate-based diets” (Animals 693141) is an original scientific paper in the broad areas of Animals journal. The subject (the test the effect of different carbohydrate sources on rumen fermentation with diet high in concentrate) is a well-known topic, but it is interesting to look into the use of a in vitro semicontinuous system.
I think the manuscript only needs very few suggestions before publication.
In general, I think that the manuscript is very well organized in terms of paragraph. The Introduction provide sufficient background and include all relevant references. The experimental design is complete. However, it is not clear if the lambs of the two groups were managed at the same way, in term of slaughtered age and weight. Why is the big difference between CI and FI for total VFA? It is not clear, the individual VFA seems similar. Please explain. The results are very well commented in the discussion section. The cited references are adequate, updated and correctly formatted according Animals guidelines.
Specific suggestions:
Keywords: include words not included in the title (i.e. sheep, citrus pulp, volatile fatty acids, microbial biodiversity…)
L13: add parenthesis before ‘Barley’
L52-54: improve English form
L93: change ‘(4, 6 and 6)’ to ‘(4, 5 and 6)’
Author Response
REVIEWER 2: Thank very much for your comments.
- As it was already stated in the former version (now in L91-101), lambs were weaned at the same age, but thereafter those given concentrate growth faster and reached higher weight than the other given forage. Therefore, time from weaning to slaughter was lower, and slaughter weight was higher, in lambs given concentrate respect to those receiving forage (35 vs. 45 days, and 20.6 vs. 16.5 kg).
- The extent of fermentation in CI is much higher than for FI. This is apparent not only on VFA but also on gas production, which also supports these results. However, molar VFA proportions are relatively similar, indicating that is the extent of fermentation rather than the fermentation pattern what is affected. Both aspects are mentioned and discussed in text
- Keywords have been changed to avoid at possible matching with those in the title.
L13: done
L52-54: this sentence has been rewritten following comments of both Reviewers.
L93: done
Round 2
Reviewer 1 Report
Thank you for the kindly response to my suggestions. However there are some minor changes that I consider that have not been accomplished.
L148-150: I can’t find the mention to the detector
L129: I can´t find the tRFLP acronym´s meaning in this line.
L173-175: You mentioned in your response letter that the statistical analysis has been rewritten for clarification, including the random effect and removing the mention to the block. However I can’t see these changes in the manuscript.
L368-370: I agree with the first part of the sentence dealing with the acidic capacity of CP and WB, but the second part is confusing since it seems that these substrates modulate medium biodiversity, and your results do not support this assumption, since only source of inoculum and donor animal had a marked effect.
Author Response
We appreciate your effort in reviewing the paper. Regarding your comments in this new review, probably we missed them from one version to another.
L148-152: this has been amended (now in L144)
L129: meaning of tRFLP has been included (now in L124)
L173-175: the statistical design has been redrafted (now in L74-176)
L368-370: I agree with your comment. For clarification, we have decided to remove the second part of the sentence (now in L381-382)
Thanks very much for your contribution to improvement to this paper.